# Ultrastructural Features of Membranous Replication Organelles Induced by Positive-Stranded RNA Viruses

**DOI:** 10.3390/cells10092407

**Published:** 2021-09-13

**Authors:** Van Nguyen-Dinh, Eva Herker

**Affiliations:** Institute of Virology, Philipps-University Marburg, 35043 Marburg, Germany; nguyendinhvan@live.com

**Keywords:** positive-strand RNA viruses, replication organelle, viral replication complex, membrane alterations, electron microscopy

## Abstract

All intracellular pathogens critically depend on host cell organelles and metabolites for successful infection and replication. One hallmark of positive-strand RNA viruses is to induce alterations of the (endo)membrane system in order to shield their double-stranded RNA replication intermediates from detection by the host cell’s surveillance systems. This spatial seclusion also allows for accruing host and viral factors and building blocks required for efficient replication of the genome and prevents access of antiviral effectors. Even though the principle is iterated by almost all positive-strand RNA viruses infecting plants and animals, the specific structure and the organellar source of membranes differs. Here, we discuss the characteristic ultrastructural features of the virus-induced membranous replication organelles in plant and animal cells and the scientific progress gained by advanced microscopy methods.

## 1. Introduction

During infection, positive-strand RNA viruses utilize the host’s cellular membranes to support every step of their replication cycle, i.e., virus entry, replication of the genome, and assembly and release of virions. These viruses induce (endo)membrane rearrangements in host cells to create a protective microenvironment for replication of their genomes and for subsequent production of new virions [1]. These endomembrane rearrangements form structures termed viral replication organelles (ROs), which are essential for virus replication. They are thought to shield viral replication intermediates from recognition and to protect them from the host cell defense systems, such as RNA silencing or interferon induction [2]. The ROs are confined membranous compartments generated by extensive alteration of (endo)membrane structures. While these membranous structures are essential for viral RNA replication, expression of single viral proteins is often enough to induce them, but size and detailed structural features may be different in the absence of virus replication.

These endomembrane re-arrangements can differ morphologically, from simple dilated membranous structures to very complex structures such as multi-vesicular bodies (Figure 1).

During the infection, viral proteins as well as hijacked host proteins target the (endo)membrane system of the host to remodel it. Through these virus-host interactions different RO structures are created, depending on the virus and the membrane source. The majority of ROs are vesicular structures. The simplest form are single membrane vesicles (SMVs), typically 50–200 nm in diameter with or without small pores that are 10–20 nm diameter which either link vesicles to each other or link the vesicle lumen to the external environment, i.e., the cytosol. These SMVs with pores, also called spherules, are believed to be generated from invaginations or evagination of the host organelle membranes. Multiple SMVs can be packed together in an organelle to form higher order vesicle packets (VPs). Slightly more complicated RO structures are double membrane vesicles (DMVs) normally ranging from 200 to 400 nm in diameter. The biogenesis process of DMVs is not well understood. DMVs can be completely sealed with two membranous layers, the outer membrane can be connected to the organelle membrane the DMV originated from, and sometimes the inner vesicles share the same outer membrane and create a larger complex of DMVs. Some DMVs have open necks linking the internal lumen of DMVs to the external environment. More complex RO organizations such as multi-vesicular bodies (MVBs) also occur. These MVB structures are big vacuoles containing numerous small disordered membranous vesicles. Other, rarer membrane alterations are multi-membrane vesicles (MMVs), which are big multi-layered membranous particles of 300–400 nm in diameter, tubule-like structures of different diameters (20–50 nm) or zippered ER. Finally, massive unstructured membranous aggregates, which are called convoluted membranes (CMs), are frequently observed in virus-infected cells.

Membrane sources are different membranous organelles such as the endoplasmic reticulum (ER), the Golgi apparatus, peroxisomes, mitochondria, and the plasma membrane, and in plants, chloroplasts, and tonoplasts. In the following sections, we discuss the ultrastructural features and membrane origins of ROs to integrate them into the cell biological context of the infected cell (Table 1).

## 2. Structure and Origin of Plant Positive-Strand RNA Virus Replication Organelles

The ROs of positive-strand RNA viruses in plants are derived from different membranous organelles of the host including the ER, Golgi, peroxisomes, chloroplasts, and tonoplasts [34,35] (Figure 2).

### 2.1. The Secretory Pathway Represents a Major Source for Membranes of Replication Organelles

The secretory pathway of plant cells is frequently targeted by viruses as a source of membranes [34]. Like its mammalian counterpart, it is composed of a complex membrane network including the ER, the Golgi apparatus, the trans-Golgi network (TGN), and endosomes. This pathway is involved in the synthesis, modification, and transport of proteins, lipids, and polysaccharides [35]. Among those organelles, the ER is most frequently targeted by viruses for their productive replication. The ER is an extensive membrane network with specialized subdomains that occupies large parts of the cell and is the prime spot for lipid and protein synthesis. Increased protein (and lipid) synthesis occurs during infection with diverse viral species and can trigger ER stress responses. 

During infection with positive-strand RNA viruses, the ER structure is often dramatically changed due to the interaction between viral and host membrane proteins to form the ROs. Some viruses, such as brome mosaic virus (BMV), tobacco mosaic virus (TMV), and red clover necrotic mosaic virus (RCNMV), induce massive ER proliferation forming ER aggregates either in the perinuclear region or randomly dispersed in the cytoplasm that are called convoluted membranes or membranous web [36,37,38,39]. Other viruses such as beet black scorch virus (BBSV) or tobacco necrosis virus (TNV-W) induce ER membrane dilations and invaginations that are rounded structures of up to 400 nm within the ER cisternae [3,40]. Along with the rearrangement of ER membranes, viruses also form higher order membrane structures called vesicle packets (VPs) containing small vesicle structures which are 50–100 nm in diameter [3]. Most of them are spherules composed of single or double membranes, called single or double membrane vesicles, SMVs or DMVs, respectively. Those vesicular structures are the areas where viruses replicate their genome [3]. The spherule structures in BBSV-infected cells are arranged along the VP membranes and are SMVs. Each spherule vesicle has a narrow neck (5–10 nm in diameter) linked to the VP membrane and thus connecting the spherule interior to the cytoplasm, suggesting that they are formed by invagination of ER membranes [3] (Figure 2B). Those VPs containing spherules with opened necks to the cytoplasm are also reported in other positive-strand RNA viruses that infect animal cells such as viruses in the *Flaviviridae* family [17,41]. 

In contrast to BBSV, the ROs of peanut clump virus (PCV) in tobacco protoplasts form VPs containing multiple SMVs which are called multivesicular bodies (MVBs). These MVBs contain multiple disordered membranous vesicles of 80–200 nm in diameter often in one row of vesicles and surrounded by a single membrane [5] (Figure 2C). Interestingly, Turnip mosaic viruses (TuMV), do not only induce formation of SMVs but also of DMV–like structures that are found in the perinuclear cytoplasmic region [4] (Figure 2A). The DMVs formed during TuMV infection occur during the late stage of infection concomitantly with massive membrane arrangements leading to altered endomembrane structures such as dilated ER and membranous inclusion bodies [4]. Cytoplasmic or membranous inclusion bodies (MIBs) were observed not only in TuMV infection but also in cells infected with different viruses such as wheat yellow mosaic virus (WYMV). WYMV forms MIBs in infected wheat plants that are large, amorphous, crystalline lattice-like inclusion bodies in the cytoplasm. The periphery of these MIBs appears to be connected to the rough ER [6] (Figure 2H), but high-resolution structural information is not available yet. 

However, other membranous structures beside spherular invaginations and vesicles can support RNA virus genome replication. For example, BMV in yeast cells can replicate the RNA at multilayer stacks of appressed double membranes [7] (Figure 2I). In infected cells, the relative expression levels and interactions between viral 1a and 2a-pol proteins can change the structure of perinuclear membrane rearrangements associated with RNA replication from small spherular invaginations to large stacks of 2–7 appressed layers of double-membrane ER. Intriguingly, these membrane stacks are highly ordered with 50–60 nm spaces, which is exactly the same width as the diameter of the spherules. These karmellae-like, multilayer structures are composed of stacks of ER that arise around the nucleus by folding over continuous sheets of ramified, double-membrane ER with its enclosed lumen. The double-membrane layers contain 1a and 2a-pol proteins and support BMV RNA replication but were not observed in yeast cells when only 1a or only 2a-pol proteins were expressed. Individual expression of BMV 1a induces only perinuclear spherules while 2a-pol alone does not cause any membrane alterations [7]. 

### 2.2. Peroxisomes and Mitochondria as Membrane Origins

The plant peroxisome is a single membrane-bound organelle that is solely responsible for beta-oxidation of fatty acids and the glyoxylate cycle, reactive oxygen species and reactive nitrogen species metabolism, and is involved pathogen defense. It is also one of the main target organelles for viruses as a membrane source to form ROs, especially for viruses in the *Tombusviridae* family such as tomato bushy stunt virus (TBSV) or cucumber necrosis virus (CNV) [9,10]. TBSV replicates in peroxisome-derived MVBs both in plant and yeast cells that are often found in close proximity to mitochondria (Figure 2G) [9]. Those MVBs are interconnected through membranes and might be nascent peroxisomes whose maturation and detachment from the ER is blocked by viral factors. In *N. tabacum* cell lines, TBSV p33 protein targets to peroxisomes and induces clustering and the formation of peroxisomal ghosts, but not MVBs, when expressed on its own [42]. CNV infection induces peroxisome biogenesis to form ROs [10]. Following infection, the peroxisomal boundary membranes are highly vesiculated, leading to the formation of doughnut- or C-shaped MVBs with the central region containing cytoplasmic material. The interiors of these doughnut-shaped MVBs contain many single-membrane vesicle-like structures with 80–150 nm in diameter. These vesicles appear to be connected to the MVB boundary membrane through a neck, and they provide the sites for CNV genome replication [10]. If peroxules that form in response to oxidative stress, which often occurs during virus infection, are hijacked by viruses as well, is currently unknown.

Interestingly, members of *Tombusviridae* not only target the ER or peroxisomes but also the mitochondria to form ROs to support viral replication as exemplified by melon necrotic spot virus (MNSV) and Carnation Italian ringspot virus (CIRV) [11,12]. In MNSV-infected cells, the mitochondrial structure is dramatically altered, and these abnormal organelles are frequently found close-by lipid droplets and ER membranes [11] (Figure 2D). Ultrastructural changes include dilated cristae and a vesiculated outer membrane. This vesiculated membrane forms multiple single-membrane vesicles with 45–50 nm in diameter which surround the large dilations inside the mitochondria. These vesicles appear to be connected to the cytoplasm or to the internal lumen of the large dilations through neck-like structures. Immuno-EM suggests that MNSV RNA and capsid proteins reside in the large dilations of abnormal mitochondria, suggesting that MNSV performs its genome replication as well as packaging in mitochondria and possibly within the interior of the vesicles [11]. 

### 2.3. The Chloroplast and Tonoplast Are Plant-Specific Membrane Sources

One organelle unique in plant cells that is also a target structure for many viruses is the chloroplast. Chloroplasts are membrane-rich organelles that conduct photosynthesis [43]. Barley stripe mosaic virus (BSMV) is a member of family *Virgaviridae* that alters chloroplast morphology during infection. In BSMV-infected plant cells, the membranes of the chloroplasts change dramatically with clusters of outer membrane-derived invaginated spherules (diameter ~50 nm with a neck of 11 nm) within inner membrane-derived packets (average diameter 112 nm) [13] (Figure 2E). The small spherules are linked via neck-like structures to the cytosol and immune-EM analysis revealed the presence of the viral RNA and replication proteins, suggesting that these spherules are the site of BSMV genome replication. In addition, big cytoplasmic invaginations surrounded by double membranes that contained virions were observed inside the chloroplasts [13]. This suggests that in addition to RNA replication, viral assembly takes place within the chloroplast.

The semipermeable membrane surrounding the vacuole is the tonoplast, an organelle that plays an important role in osmotic regulation of turgor pressure and that is targeted by viral infection. Already in the 1980s, cucumber mosaic virus (CMV)-infected leaf cells were shown to harbor tonoplast-associated vesicular structures [44]. The latest findings revealed that vacuole membranes are remodeled and invaginated in cells infected with CMV or tobacco necrosis virus A Chinese isolate (TNV-AC) [14]. Membrane invaginations form spherules at the periphery of the vacuole that are 50–70 nm in diameter (Figure 2F). These spherules contain neck-like structures that connect their interior with the cytosol. Interestingly, in CMV-infected cells, besides the spherules located at the tonoplast membrane, peripheral spherule-containing MVBs were also observed. The spherules inside the MVBs are also open towards the cytoplasm with a neck-like structure and the interior of the MVB seems to be connected to the vacuole. In addition to spherule-containing MVBs, membrane compartments harboring viral particles are found in close proximity to the vacuole and the ROs [14]. 

## 3. Structure and Origin of Animal Positive-Strand RNA Virus Replication Organelles

Similar to plant viruses, genome replication of all positive-strand RNA viruses that infect animal cells is intimately associated with membranes. The viral ROs supporting the replication of the viral genomes are generated from different host cellular membranous organelles including the endoplasmic reticulum (ER), the Golgi apparatus, mitochondria, lysosomes, and the plasma membrane [1] (Figure 3). 

### 3.1. The ER Is the Main Hub for Animal Virus RO Fomation

Among the different membrane-bounded organelles, the ER represents the main membrane source for many positive-strand RNA virus ROs in animal cells [46]. The *Flaviviridae* family is one positive-strand RNA virus family that is well-known for ER-based RO formation [16,46,47]. In cells infected with dengue virus (DENV) tick-borne encephalitis virus (TBEV), West Nile virus (WNV), or Zika virus (ZIKV), the ER structure is dramatically altered owing to viral genome translation and replication. These viruses induce the formation of different membranous structures in the cytoplasm: vesicle packets (VPs) inside the ER, convoluted membranes (CMs) (Figure 3B), which are peculiar membranous aggregates with unknown function [18,47,48], and, in some cases, dilated ER, which are enlarged rough ER cisternae filled with granular material e.g. in TBEV infected cells [41,49]. In Hela cells transfected with a TBEV DNA replicon, the dilated ER cisterna grow to big cytoplasmic vacuoles containing small spherule-like structures 80–100 nm in diameter, which have open necks towards the cytoplasm (Figure 3A) [41]. The most prominent membranous structures derived from the ER in flavivirus-infected cells are the VPs that are the sites of viral genome replication and thus represent the ROs [17,41,50]. Early immuno-EM studies in DENV-infected insect cells indicated that VPs (or smooth membrane structures, SMS) are the site of DENV RNA replication [50]. These VPs are ER-derived membranous structures that are dilated ER cisterna containing single-membrane vesicles (SMVs) with a diameter of 80–150 nm [15] (Figure 3C). These SMVs originate from the invagination of the ER membrane into the ER lumen, have a spherule structure with small, 10–15 nm diameter necks opening to the cytoplasm. Necks were also observed linking SMVs inside the VPs in WNV-infected cells [16]. Densely packed viral particles are frequently within the ER in close proximity to VPs [15] (Figure 3C). Interestingly, an electron tomography (ET) study of TBEV-infected human neuronal cells investigated the proliferating ER in infected cells and found additional tubule-like structures of different diameters (20–50 nm) inside ER cisternae [19] (Figure 3D). In some instances, these tubule-like structures have direct contacts with viral particles inside these proliferated ER cisterna [19]. The function of these tubule-like structures is thus far unknown; they may represent membranous structures involved in viral replication, abnormal cellular structures arising due to altered membrane metabolism, or a feature of cellular process to limit the viral infection [51]. 

Among the members of *Flaviviridae*, hepatitis C virus (HCV) is somewhat unique regarding the prototypical RO structures. In HCV-infected hepatocyte cells, ER membranes are intensively rearranged to form the membranous web (MW). The MW contains vesicles of different morphologies, mainly SMVs or DMVs, embedded in a matrix of membranes which are sometimes close to or wrap tightly around lipid droplets [21,22]. HCV infection as well as expression of single HCV proteins induce different types of membranous vesicles in cells [20,21,22]. While NS3/4A and NS4B induces only SMVs, NS5A induces MMVs and infrequently DMVs [22]. However, expression of the complete replicase complex (NS3-NS5B) is needed for formation of DMVs that are indistinguishable from the ones observed in infection [22]. In HCV-infected cells there are vesicles in clusters containing SMVs of variable sizes (100–200 nm in diameter), sometimes sticking together and harboring internal invaginations, and SMVs of a homogeneous size (~100 nm in diameter) that are clustered together and sometimes arrayed around lipid droplets. However, the most prominent vesicular structure induced by HCV are DMVs, likely representing the ROs. The DMVs are heterogeneous in size, with an average diameter of 200-400 nm, and are morphologically similar to membrane alterations identified in cells infected with coronaviruses [23] or picornaviruses [27]. These vesicles are characterized by two closely apposed membranes. EM/ET analysis revealed that most of the DMVs are generated from the ER and some of them are still connected to ER sheets via their outer membrane [22]. Although most of DMVs are completely closed structures and it is still unknown why HCV would induce these closed structures, a small percentage of them (8–10%) [22] has an opening neck towards the cytosol. The opened and closed DMVs thus may reflect the different stages of DMV “maturation”, early and late, respectively [22]. An immunolabeling study of purified DMVs revealed an enrichment for viral proteins as well as dsRNA suggesting that DMVs indeed play an important role for viral RNA replication [52]. Viral RNA amplification may occur inside DMVs, which would allow the exit of newly synthetized viral genomes as long as the DMV is open, but replication might also occur on the outer surface of DMVs [22,52]. A more recent study using correlative light and electron microscopy (CLEM) indicated that DMVs emerge from ER membranes which are tightly wrapped around lipid droplets [45] (Figure 3E). EM/ET analysis of HCV-infected cell revealed two types of lipid droplets: lipid droplets that are tightly wrapped by the ER and that stain positive for the HCV glycoprotein E2 and nonstructural protein NS5A by immunofluorescence microscopy as well as lipid droplets that are not wrapped by ER and that do not stain positive for E2 and NS5A. These data suggest that HCV proteins trigger wrapping of ER membranes around lipid droplets. This tightly closed contact between DMVs and ER-wrapped lipid droplets may enable short-distance trafficking of viral RNA from replication vesicles to assembly sites at lipid droplet–associated ER membranes [45]. Later during HCV infection, multi-membrane vesicles (MMVs) with an average diameter 350–400 nm are generated, likely originating from DMVs through secondary enwrapping events [22]. 

DMVs are observed not only in HCV infection but also during infection with other positive-strand RNA viruses, such as members of *Nidovirales*, including coronaviruses and arteriviruses [1]. DMVs are well-known typical ROs of coronaviruses [23,24]. A new study employing 3D reconstructions using FIB-SEM (focused ion beam milling combined with scanning EM) to determine morphological alterations induced in severe acute respiratory syndrome coronavirus 2 (SARS-CoV-2)-infected human lung epithelial cells revealed extensive fragmentation of the Golgi apparatus, alteration of the mitochondrial network, and recruitment of peroxisomes to viral ROs, which are clusters of DMVs [25]. In the SARS-CoV-2-infected cells, the ER network was altered intensively to generate the ROs, which consist predominantly of DMVs with an average diameter of 250–350 nm. Theses DMVs were tightly connected with the ER network linking the outer membrane to ER-derived structures such as ER connectors. Similar to DMVs in HCV-infected cells, the DMVs in SARS-CoV-2-infected cells are mostly closed DMV structures. However, DMV-DMV contacts were observed in SARS-CoV-2-infected cells, either through funnel-like junctions between two DMVs or fused DMVs consisting of multiple vesicles sharing the same outer membrane [25]. As described above, SARS-CoV-2 induces the formation ER connectors between the DMVs and ER tubules [25] (Figure 3F). These membranous structures were also described as zippered ER in gamma- or betacoronaviruses, such as infectious bronchitis virus (IBV) or Middle East respiratory syndrome coronavirus (MERS-CoV) [24,26]. The zippered ER or ER connectors lack luminal space, suggesting that they are formed through zippering or collapsing of ER cisternae. However, in contrast to SARS-CoV-2, electron tomograms showed that IBV-induced spherules are tethered to zippered ER and that there is a small pore connecting the interior of the spherule with the cytoplasm [26]. Of note, in a recent study of ZIKV, zippered ER structures were also observed in infected cells [17]. 3D reconstruction of regions containing zippered ER in ZIKV-infected cells revealed that the collapsed ER was connected to regions containing invaginated replication vesicles [17,47]. 

### 3.2. Further Down the Secretory Route, the Golgi Apparatus Supports RO Formation

Many viruses rely on the secretory route through the Golgi apparatus for maturation and release of viral progeny. However, some viruses also employ Golgi membranes to establish their ROs for viral RNA replication, e.g. poliovirus or coxsackieviruses, which are members of *Picornaviridae* family. Membrane alterations in poliovirus-infected cells include the formation of SMVs and DMVs [28]. A recent publication employing immuno-EM with subsequent diaminobenzidine (DAB) labeling suggested that membrane rearrangements in poliovirus-infected cells may occur in a sequential manner [27] (Figure 3J). In the early stage of infection, small clusters of SMVs appear. Later in infection, they are replaced by either round or irregularly shaped DMVs. Interestingly, the small clusters of SMVs of 100–200 nm in diameter strongly stained positive for a Golgi antigen, GM130, a cis-Golgi marker, but not for calnexin, an ER marker. These data suggest that the ROs of polioviruses may originate from the Golgi apparatus. However, it is too early to exclude a role of the ER for biogenesis of these ROs as ER-proteins might be dislocated during RO formation. dsRNA, i.e. viral RNA replication intermediates, as well as metabolically labeled viral RNA were detected in both SMVs and DMVs of poliovirus ROs, suggesting that both structures are relevant sites for poliovirus RNA synthesis [27]. 

### 3.3. Mitochondria, Lysosomes, and the Plasma Membrane Are Involved in RO Formation

Interestingly, the flock house virus (FHV), a member of the family *Nodaviridae* targets the mitochondria to form ROs supporting their RNA replication. In FHV-infected *Drosophila* cells, the mitochondrial outer membrane is dramatically altered [29] (Figure 3G). The virus induces the formation of invaginations at the outer mitochondrial membrane into the spherule structures with an average diameter of 50 nm. All spherules are outer mitochondrial membrane invaginations with their lumen connected to the cytoplasm through a small pore of 10 nm in diameter, which is sufficient for ribonucleotide import and product RNA export [29]. A recent cryo-electron tomography study showed the presence of electron-dense structures within the spherules, which likely corresponds to the viral RNA as the volume correlated well with viral RNA length [30] (Figure 3H). This study additionally revealed the structure and symmetry of the proteins that form the pore complex. These pore complexes were frequently associated with long cytoplasmic electron-dense trails, likely representing exported viral RNA [30].

The lysosome is another cellular organelle which is a favorite target for some positive-strand RNA viruses such as rubella virus (RUBV) and members of *Togaviridae*, including Semliki Forest virus (SFV) and sindbis virus (SINV) [53,54,55]. These viruses alter lysosome and endosome structures to form cytopathic vacuoles (CPVs) that represent the viral ROs [54]. In RUBV-infected cells, the rough ER, mitochondria, and the Golgi are clustered around CPVs, which are linked to the cytosol and enclose vesicular structures [31] (Figure 3I). These organelles contain active ROs from which replicated RNA is transported to virion assembly sites at Golgi membranes. These CPVs have a quite variable diameter of 600–2000 nm. Electron tomography and 3D reconstruction revealed that CPVs enclose a variety of different membrane structures such as stacked membranes, rigid membrane sheets, small vesicles, and larger vacuoles that are connected through membrane contacts with each other and functionally connected to the endocytic pathway. CPVs have additional membrane contact sites to other cellular organelles such as the rough ER and Golgi vesicles, but not to nearby mitochondria. Immunogold labeling confirmed the presence of replicase complex proteins and dsRNA inside CPVs, suggesting that RNA synthesis occurs on or in vesicles within the CPVs [31]. 

As mentioned above, alphaviruses, such as SFV, SINV and WEEV, are known to induce formation of CPVs in infected cells, which are modified lysosomes and endosomes and the sites of viral RNA replication. Interestingly, in SINV-infected cells spherules containing dsRNA and nonstructural protein (nsP) are initially formed at the plasma membrane [32,33]. Immunofluorescence microscopy and EM revealed that at early times of infection, viral nsPs as well as dsRNA replication intermediates locate to spherules the plasma membrane [33] (Figure 3K). These spherules form as evaginations at the plasma membrane and the presence of plasma membrane-associated dsRNA and ns proteins suggest that they represent ROs. Later in infection, these spherules are internalized by endocytosis; trafficking and maturation to CPVs is dependent on phosphatidylinositol 3-kinase activity and the cytoskeleton [33], highlighting the often complex nature of viral RO formation.

## 4. Recent Technical Developments and Challenges

For multiplication, viruses need to infect a suitable host cell to be able to replicate their genome, to produce and release new infectious virions, and thus continue the next round of the infectious cycle. The interactions of viruses with their hosts are highly dynamic, diverse and complex, and occur on multiple levels. It is important to elucidate the molecular mechanisms of these virus-host interactions in order to understand virus replication cycles and how viruses affect and alter the cell biology of their host to support viral replication. This knowledge is not only important for better understanding of the biology of viruses but also to support control of viral infections, to predict their effect on ecology and human health, and to design effective antiviral strategies against chronic and emerging viral infectious diseases. 

“Seeing is believing”, we clearly trust observations that we can visualize. Microscopy, especially high-resolution light/fluorescence and electron microscopy (EM) are important tools for visualizing structures of viral and host cell components and thus for the generation of general concepts governing virus-host interactions. Indeed, EM and virus research developments are deeply intertwined since the invention of EM [56,57]. EM is one of the critical methods to elucidate how viruses replicate in the microstructure environments of the infected cell in order to produce new virions [58]. In general, EM techniques encompass two main applications: transmission EM (TEM) and scanning EM (SEM), which each are different microscopic techniques [59]. The resolution of SEM is lower than that of TEM. In contrast, SEM provides a larger sample scanning ability or a bigger field of view for both surface and volume. Therefore, TEM is the favorite method to study small structures in detail, whereas SEM applications help to expand the sample scales.

The combination of EM with advanced light microscopy techniques termed correlative light and electron microscopy (CLEM) provides even more detailed information as it allows to analyze the dynamics and localization of viral and/or host protein-protein interactions in the context of detailed structural aspects of the intracellular environment. In this method proteins are visualized through fluorescent tags or antibodies using light microscopy in order to find rare biological events or to identify specific structures prior to characterizing the structures and their surroundings at high-resolution using EM. The current full spectrum of state-of-the-art microscopic techniques covers an extensive range of scales, resolutions, and information. Many of the methods mentioned together with the viral RO structures in this review, such as electron (cryo)tomography, CLEM, volume SEM, or 3D TEM have thrived and were further advanced within only two decades, especially since cryo-EM was discovered and developed in the 1980s [60].

The newly advanced electron tomography (ET), including volume SEM and cryotomography, has been a useful method in elucidating the 3D volume architecture of viral ROs. Volume SEMs such as serial block face SEM and focused ion beam milling (FIB)-SEM have been used to explore virus-host interaction with the nanometer resolution in wider and thicker volume samples including tissues. Furthermore, advanced cryo-FIB-SEM techniques are applied on cryo-stage specimens, which can help to avoid the artifacts of conventional EM sample preparation due to chemical fixation and staining processes and can also help to improve the stabilization of native structures in the specimen [61]. Although currently cryotomography of FIB-milled cryo-lamellae is the outstanding method in ET, the area that can be investigated is restricted to a very small and thin cellular region (the cryo-lamella) [62]. Difficulties in sample preparation combined with the need for highly demanding technical skills and high equipment costs are further limitations that are needed to be solved with technology developments in the future [62]. On the contrary to volume SEM, cryotomography methods can yield magnificent structural details with molecular-level resolution of the viral ROs in the cryo-native condition [63]. Cryotomography is currently one of the most powerful methods for investigation and characterization of the biological structures of viral ROs from the macro-structural morphology to the nano-organization of detailed protein structures which were presented in many current studies on viral ROs discussed in this review. Furthermore, current cryo-CLEM application, which combines cryo-light microscopy and cryo-EM opens a new way in investigating the molecular mechanisms of virus-host interactions more specifically and more accurate under cryogenic conditions [63]. However, similar to cryo-FIB-SEM, only a small area of the targeted cellular structure can be processed for investigation and the processing of cryotomography requires highly developed technical skills, limiting the popularity of 3D-cryoEM. For cryo-CLEM, the limited resolution of cryo-light microscopy, mostly based on wide-field light microscope also decreases the accuracy of this technique when it comes to localization of specific structural protein or events [64].

Of course, one main obstacle when investigating virus-infected specimens is the need for inactivation, especially for human pathogenic viruses. Thus, these samples require strong fixation that may cause artifacts. Alternatively, all steps including the image acquisition under cryo conditions have to be performed under biosafety containment, which is difficult to implement. Thus, we may need to rely on non-pathogenic model viruses for some of the advanced microscopy techniques. 

## 5. Conclusions

Positive-stranded RNA viruses dramatically remodel intracellular membranes into distinct RO structures that support the synthesis of viral RNA. ROs provide optimal micro-environments for viral genome replication and shield replication intermediates such as double-stranded RNA (dsRNA) from detection by innate immune sensors. Many questions about the biogenesis process viral ROs remain unanswered, i.e., for many viruses we do not have detailed information on host factors such as proteins and specific lipids that contribute to RO formation. Likewise, the dynamic nature of how and where and when during infection viral proteins required for RO formation interact with host proteins to remodel intracellular membranes into viral ROs and to stabilize the RO morphology remains to be determined. For many viruses pores connecting the RO interieur with the cytosol are observed but how viral proteins interact with host membrane proteins to stabilize these pore structures are still poorly understood [65]. The crown-shaped molecular complexes of some of the pores unveiled in recent studies of positive-stranded RNA viruses have provided us an overview of the protein complex organization of these pores [30,66,67]. However, how flexible the pores are and how the pore proteins regulate the transit of proteins and nucleotides/viral RNA from and to ROs and, possibly, coordinate it with other processes in the viral replication cycle is still poorly investigated. For other viruses, closed ROs have been observed frequently. If they are inactive/old ROs or just open up intermittently is still unclear. Elucidating how the viral replication complexes work on a molecular level and integrating biochemical knowledge with structural information gained by EM analysis are challenging goals for the future. 

## Figures and Tables

**Figure 1 cells-10-02407-f001:**
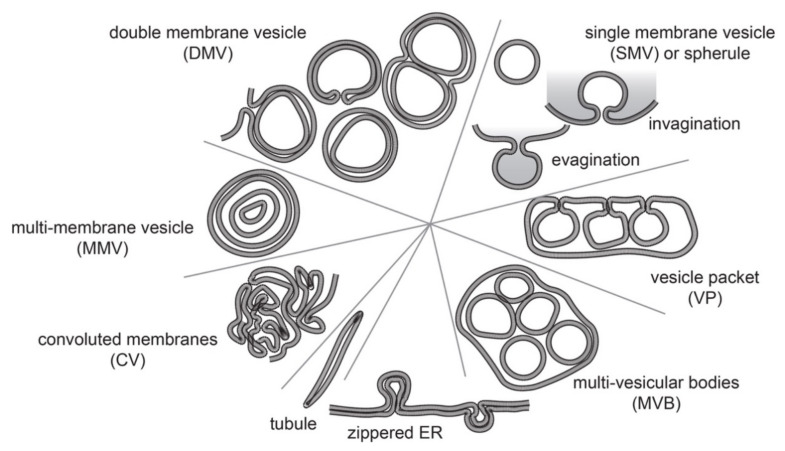
Membranous structures that occur in cell infected with positive-strand RNA viruses. Depicted are the most common membranous structures.

**Figure 2 cells-10-02407-f002:**
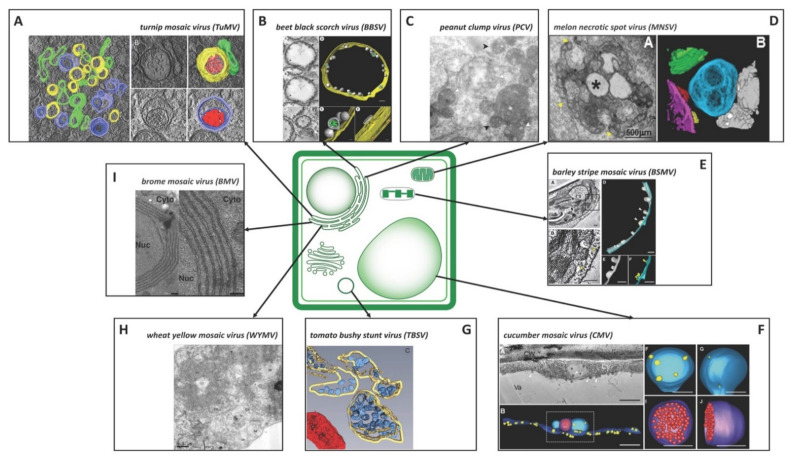
**Structure and origin of plant positive-strand RNA virus replication organelles.** (**A**) 3D architecture of TuMV-induced complex membrane structures. Overview of a single slice of a tomogram of a TuMV-infected vascular parenchymal cell. (upper right) The 3D model shows a SMV with fibrillar material inside and with an adjacent intermediate tubular structure. (lower right) 3D model of a DMV with a core of electron-dense materials. Yellow, SMVs; light red, electron-dense materials; green, intermediate tubular structures; light blue, outer membranes of DMVs; dark blue, inner membranes of DMVs; dark red, the electron-dense materials inside DMVs [4]. (**B**) Dilated ER of BBSV-infected cells with SMVs (left) and 3D surface reconstruction of the tomogram corresponding to the intact spherules (right) depicting the outer ER membrane (yellow), BBSV-induced spherules (gray), and fibrillar materials inside the spherules (green). Scale bars 100 nm [3]. (**C**) Electron microscopy of MVB structures in PCV-infected BY-2 protoplasts. White arrows indicate clusters of vesicles. Single arrowheads correspond to MVB; MVB containing disordered membranous vesicles are indicated by black arrowheads, whereas those containing one row of vesicles that are surrounded by a single membrane are indicated by white arrowheads. White asterisks correspond to electron-dense material without detectable vesicles [5]. (**D**) TEM analysis and 3D reconstruction of MNSV-induced altered mitochondria. (left) TEM image of altered mitochondria. Numerous vesicles were observed on the external surface as well as internal large invaginations and internal dilations (star), or both. Yellow arrowheads indicate the pores connecting the lumen of the dilation to the surrounding cytoplasm. (right) 3D model of MNSV-induced altered mitochondria (blue, yellow, red, and purple) with large dilations inside and close interactions with lipid droplets (grey) and chloroplasts (green) [11]. (**E**) BSMV-induced chloroplast membrane rearrangement and 3D model of altered chloroplast membranes. (left) Tomogram slices of altered chloroplast membranes from leaves of BSMV-infected *N. benthamiana*. The arrowheads indicate the same spherules in different slices. (right) 3D model of remodeled chloroplast membranes induced by BSMV indicating the outer chloroplast membrane (cyan), inner chloroplast membrane (gray), and spherules derived from the outer membrane (yellow) [13]. (**F**) 3D visualization of remodeled tonoplasts in CMV-infected cells. (upper left) Tomogram slice of a CMV-infected *N. benthamiana* leaf cell. CMV-induced spherules are observed on a vacuolar membrane and in a MVB (arrowheads). The cell wall (CW), cytosol (Cy), and vacuole (Va) are indicated. Scale bar 500 nm. (lower left) 3D model depicting the vacuolar membrane (dark blue), MVBs (light blue), spherules on the vacuolar membrane and in the MVBs (yellow), and a membrane compartment (purple) with virus particles (red). (upper left) 3D model of the MVB with spherules open to the cytosol. (lower left) 3D model of the membrane compartment with virus particles. Scale bars 200 nm [14]. (**G**) 3D reconstruction of TBSV ROs in wild-type yeast cells characterized by peroxisome-peripheral MVBs depicting the MVB membranes (yellow), vesicle-like spherules (blue) located close to a mitochondrion (red) [9]. (**H**) Electron micrographs of the mesophyll cells of WYMV-infected wheat. The presence of membranous inclusion body structures in the cytoplasm. The ER, membranous inclusion (MI), mitochondria (Mt), pinwheel inclusion (PW), and virus particles (VP) are labelled [6]. (**I**) A series of 2–7 appressed layers of double-membrane ER in yeast cells expressing both 2a pol and 1a of BMV, double-membrane ER layers are separated by regular, 50–60-nm spaces, the nucleus (Nuc) and cytoplasm (Cyto) are indicated. Scale bars 100 nm [7] Copyright (2004) National Academy of Sciences, U.S.A. The different parts were reproduced with permission.

**Figure 3 cells-10-02407-f003:**
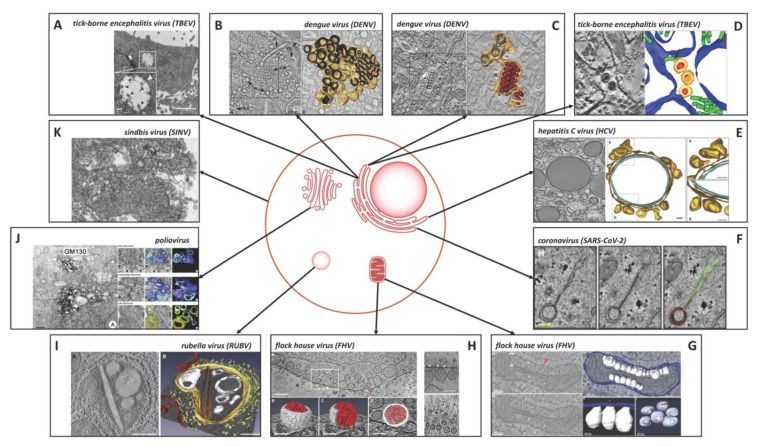
**Structure and origin of animal positive-strand RNA virus replication organelles.** (**A**) TEM images of HeLa cells transfected with the TBEV DNA replicon. White arrowheads show dilated ER areas; black arrowheads denote replication-vesicle-like structures inside the dilated ER areas. Insets show magnifications of the indicated areas. Scale bars 1 μm [41]. (**B**,**C**) DENV-infected Huh7 cells. (left) Tomogram slice shows DENV-induced convoluted membranes (CM), vesicles (Ve), and tubes (T) that form a network of interconnected membranes in continuity with ER membranes. (right) 3D surface model of the membranes in the boxed area. The outer (cytosolic) face of the continuous membrane network is depicted in yellow; the ER lumen is dark [15]. (left) Stacked virus particles are in ER cisternae that are directly connected to virus-induced vesicles (white arrow). (right) 3D surface model of the virus-induced structures in the boxed area showing the continuity of virus-and vesicle-containing ER cisternae. ER membranes are depicted in yellow, inner vesicle membranes in light brown, and virus particles in red [15]. (**D**) Proliferation of the ER in human neuronal cells infected with TBEV. TBEV particles and TBEV-induced vesicles are located inside the proliferated and reorganized cisternae of the rough ER. 3D reconstruction of lamellar whorls, which are surrounded by cisternae arising from the rough ER (blue) and accommodate tubule-like structures (green). Detailed image shows the connection between the envelope (yellow) of a TBEV particle with nucleocapsid (red) and a tubule-like structure (indicated with an arrow) inside the rough ER. Scale bars 50 nm [19]. (**E**) 3D model of the HCV replication organelles surrounding lipid droplets. Electron tomography suggests that DMVs arise from ER membranes that are tightly wrapped around lipid droplets. (Left) Single tomographic slice of an HCV-infected cell with lipid droplets that are tightly wrapped by ER membranes and that stain positive for E2 and NS5A as revealed by fluorescence microscopy (not shown). (right) 3D reconstruction of the membranes surrounding the lipid droplet. ER membranes and DMVs are shown in yellow; the phospholipid monolayer of the lipid droplet monolayer membrane is shown in cyan. Insets illustrate that the DMVs originate from the wrapping ER membrane. Scale bars 100 nm [45]. (**F**) High-resolution analysis of ER-DMV interconnectivity in SARS-CoV-2-infected Calu-3 cells. Tomogram slices depict a membrane connector or zippered ER (light green) in contact with a DMV (red). (right) Superposition of rendered DMV and ER. Scale bars 200 nm [25]. (**G**) Tomogram slices and 3D reconstructions of mitochondria in FHV-infected *Drosophila* cells. (Left) Tomogram slices showing FHV-induced spherule rearrangements of a mitochondrion. Labels denote outer mitochondrial membrane (OM) and inner mitochondrial membrane (IM). White arrowheads indicate the necks that connect spherules to the OM. Asterisks mark two spherules that connect via necks to the OM. A red arrow marks the ∼10 nm channel connecting the spherule interior to the cytoplasm. (upper right) 3D tomogram image with blue indicates OM, white indicates FHV spherules. (lower right) A close-up view of the connections between the OM and the spherules and 90° rotation of spherules showing the channels that connect the spherule interiors to the cytoplasm [29]. (**H**) (upper left) Tomogram slice of FHV spherules in a mitochondrion. Mitochondrial outer membrane (red), spherule membrane (blue), interior spherule filaments (black), and spherule openings (white) are indicated with arrowheads. Scale bar 100 nm. (lower left) 3D reconstruction of the spherule outlined in upper panel. Scale bars 50 nm. (right) Filaments are associated with FHV spherule pores. Tomographic slices with arrowheads pointing to the mitochondrial outer membrane (red), the spherule membrane (blue), the spherule opening (white), and the extruding filaments that likely represent viral RNA) (black). Scale bars 100 nm [30]. (**I**) 3D ET volumes of RUBV replication complex in BHK-21 cell. Tomogram slice (left) and the corresponding 3D model (right) of a CPV (yellow) surrounded by the rough ER (light green) and containing a number of vacuoles, vesicles, and a rigid straight sheet (brown) that is connected with the periphery of the CPV; mitochondria (red), vesicles and vacuoles (white) and cytoplasm (grey). Scale bars 200 nm [31]. (**J**) Poliovirus ROs in HeLa cells. (left) Viral replication structures are strongly associated with staining for a Golgi antigen, GM130. Scale bar 500 nm. (right) 3D reconstructions of poliovirus ROs at the early, intermediate, and late stages, 3, 4, and 7 hours post infection, respectively, each depicting central slices in tomographic volumes, central slices with segmented overlays, and segmented volumes, with blue indicating SMVs and yellow and green indicating inner and outer membranes of DMVs, respectively. Scale bars 100 nm [27]. (**K**) Plasma membrane invaginations and vacuole formation in SINV-infected BHK-21 cells. Scale bar 200 nm [32]. The different parts were reproduced with permission.

**Table 1 cells-10-02407-t001:** Membrane sources and morphologies of the replication organelles (ROs).

	Membrane Source	Replication Organelles (RO)		Virus	Virus Family	Ref.
**plant viruses**	endoplasmic reticulum (ER)	vesicle/spherule	single membrane	beet black scorch virus (BBSV)	*Tombusviridae*	[3]
			double membrane	turnip mosaic virus (TuMV)	*Potyviridae*	[4]
		multi-vesicular body		peanut clump virus (PCV)	*Virgaviridae*	[5]
		membranous inclusion body		wheat yellow mosaic virus (WYMV)	*Potyviridae*	[6]
		appressed double-membrane layers		brome mosaic virus (BMV)	*Bromoviridae*	[7]
	Golgi	dilated Golgi		tomato spotted wilt virus (TSWV)	*Tospoviridae*	[8]
	peroxisomes	multi-vesicular body		tomato bushy stunt virus (TBSV), cucumber necrosis virus (CNV)	*Tombusviridae*	[9,10]
	mitochondria	multi-vesicular body		melon necrotic spot virus (MNSV), Carnation Italian ringspot virus (CIRV)	*Tombusviridae*	[11,12]
	chloroplast	single membrane vesicle/spherule		barley stripe mosaic virus (BSMV)	*Virgaviridae*	[13]
	tonoplast	single membrane vesicle/spherule		cucumber mosaic virus (CMV)	*Bromoviridae*	[14]
				tobacco Necrosis Virus-Serotype A (TNV-A)	*Tombusviridae*	[14]
**animal viruses**	endoplasmic reticulum (ER)	convoluted membrane		dengue virus (DENV), West Nil virus (WNV), Zika virus (ZIKV), tick-borne encephalitis virus (TBEV)	*Flaviviridae*	[15,16,17,18]
		vesicle/spherule	single membrane			
			tubule-like structure	tick-borne encephalitis virus (TBEV)	*Flaviviridae*	[19]
		double membrane vesicle		hepatitis C virus (HCV)	*Flaviviridae*	[20,21,22]
		zippered ER		severe acute respiratory syndrome coronavirus (SARS-CoV), middle east respiratory syndrome coronavirus (MERS-CoV), SARS-CoV2, infectious bronchitis virus (IBV)	*Coronaviridae*	[23,24,25,26]
				Zika virus (ZIKV)	*Flaviviridae*	[17]
	Golgi	single and double membrane vesicle		polio virus (PV)	*Picornaviridae*	[27,28]
	mitochondria	single membrane vesicle/spherule		flock house virus (FHV)	*Nodaviridae*	[29,30]
	lysosome	cytopathic vacuole, single membrane vesicle/spherule		rubella virus (RUBV)	*Matonaviridae*	[31]
	plasma membrane	evagination, single membrane vesicle/spherule		sindbis virus (SINV)	*Togaviridae*	[32,33]

## Data Availability

Not applicable.

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
