# Peer review of "Ultrastructural Features of Membranous Replication Organelles Induced by Positive-Stranded RNA Viruses"

_cells, 2021, doi:10.3390/cells10092407_

Round 1

Reviewer 1 Report

The manuscript by Nguyen-Dinh and Herker nicely summarized ultrastructure of ROs of (+)RNA viruses. Description of ROs of each individual viruses was well-done and some most recent ones published in 2021 were also included.

Major issues:

  1. The organization of this manuscript needs a lot of improvement. It seems that the two authors, one did on plant and the other on animal viruses, just put their parts together. Are their shared features when different viruses replicate in the same organelle? What are major differences between plant and animal virus ROs in the same organelle?

  1. A very short introduction and then jumps to individual viruses. Maybe a longer introduction on organelle features or a summary of sheard features of SMV or DMV may orient readers general information of ROs.

  1. An image showing the membranous rearrangements associated with distinct ROs should be added.

Minors:

The conclusion that tonoplasts as CMV replication site has been reported in 1980s (JGV 1981, 53:343-346). These papers should be cited.

L139: the abbreviation of cytoplasmic invaginations as CIs might not be a good idea because one plant potyviral replication protein is named CI (cytoplasmic inclusion body).

L281: for FHV, a recent paper (DOI: 10.7554/eLife.25940), which is the most advanced study on viral ROs so far, should be discussed thoroughly.

L283: You may double-check that TUBV is no longer in the Togaviridae anymore but in a family of its own.   

Reviewer 2 Report

The manuscript was very well written. The breadth of viruses covered in the review with respect to the site of virus replication range from plant to and animal viruses are commendable. Great job and well put together. 

Reviewer 3 Report

The review titled „ultrastructural features of membranous replication organelles induced by positive-stranded RNA viruses“ by Nguyen-Dinh and Herker gives a very nice and very comprehensive overview of the different membrane rearrangements induced by plus-strand RNA viruses to support their replication. Overall the article is well written, concise and clearly structured. Nevertheless, it might be improved through the addition of details allowing a) easy understanding even for readers not familiar with every virus mentioned in the paper and b) a better insight into the current knowledge as to the formation of the ROs.

Specific comments:

  • While the figures are very nicely presented, I would have liked an additional table showing virus name, family and organelle it derives its RO from.
  • in most cases the membrane rearrangements are due to viral proteins and will happen in the absence of viral RNA, which is worthwhile mentioning.
  • line 86 to 96 might be expanded to explicitly tell the reader that
  1. 1a and 2apol are viral proteins not yeast proteins
  2. 1a alone can induce perinuclear spherules
  3. 2apol alone does not induce membrane rearrangements
  4. All three are implicit, but not explicitly mentioned.
  • Line 105 reference 20 might not be the best to cite for the peroxisome-origin of the MVB as the MVBs in that article are probably ER-derived due to the use of a knock-out yeast strain. Might be worthwhile to add a 2nd reference and maybe expand a bit on the role of p33 in membrane rearrangements?
  • Line 198/199 which proteins, induce which type of vesicle? This might be especially interesting in view of the discussion on DMV a bit further on in that paragraph
  • Line 39: typo: “replocation” organelles
  • Line 367: change “However, similar to cryo-FIB-SEM, only small area“ to “However, similar to cryo-FIB-SEM, only a small area“
  • Line 394/395: Improve this sentence: “However how flexible of these viral pore proteins to regulate the transit from/to ROs and, possibly, to coordinate it with other processes in the viral replication cycle is still poorly investigated”
